# The Competitive Edge: T6SS-Mediated Interference Competition by *Vibrionaceae* Across Marine Ecological Niches

**DOI:** 10.3390/microorganisms13061370

**Published:** 2025-06-12

**Authors:** Perla Jazmin Gonzalez Moreno, Michele K. Nishiguchi

**Affiliations:** Department of Molecular and Cell Biology, University of California, Merced, CA 95343, USA; pgonzalezmoreno@ucmerced.edu

**Keywords:** antagonism, interference competition, type VI secretion system, *Vibrionaceae*, intraspecific competition, interspecific competition, microbial communities

## Abstract

Interference competition, wherein bacteria actively antagonize and damage their microbial neighbors, is a key ecological strategy governing microbial community structure and composition. To gain a competitive edge, bacteria can deploy a diverse array of antimicrobial weapons—ranging from diffusible toxins to contact-mediated systems in order to eliminate their bacterial rivals. Among Gram-negative bacteria, the type VI secretion system (T6SS) has emerged as a potent and sophisticated contact-dependent mechanism that enables the delivery of toxic cargo into neighboring cells, thereby promoting the colonization and dominance of a bacterial taxon within an ecological niche. In this review, we examine the ecological significance of T6SS-mediated interference competition by members of the *Vibrionaceae* family across a range of marine habitats that include free-living microbial communities and host-associated niches such as coral and squid symbioses. Additionally, we explore the ecological impact of T6SS-mediated competition in modulating biofilm community structure and promoting horizontal gene transfer within those complex microbial populations. Together, these insights underscore the ecological versatility of the T6SS and emphasize its role in driving antagonistic bacterial interactions and shaping microbial community dynamics within marine ecosystems.

## 1. Introduction

In the marine environment, bacteria are often in close proximity to other microbial neighbors that may also utilize the same nutrient sources or colonize the same ecological niche. As such, bacterial neighbors must either find a way to coexist or engage in antagonistic behaviors to ensure their dominance within their spatial niche [1,2,3]. These antagonistic interactions are often critical for the survival and continued persistence of distinct bacterial groups across various marine ecosystems [4]. To outcompete potential rivals, bacteria can engage in two different types of antagonistic behaviors: exploitative and interference competition. Exploitative competition refers to an indirect form of competition in which a single bacterial species or strain depletes a limited nutrient supply, thereby hindering the accessibility of this resource to other competing strains [3,5]. In contrast, interference competition is a direct form of competition where bacterial cells actively damage one another via chemical warfare or prokaryotic killing systems [3,5].

When engaged in direct interference competition, bacteria can deploy various types of intra- and interspecific antibacterial weapons to defeat prokaryotic opponents [3,6]. These antibacterial weapons are widely distributed across microbial taxa and can vary in terms of their mechanism of action, mode of target penetration, and specificity against potential targets [3]. However, most bacterial weapons can be categorized into two distinct groups: diffusible toxins or contact-dependent mechanisms. Diffusible toxins are secreted into the surrounding environment, allowing bacteria to inhibit or kill a broader range of microbial rivals within a large spatial distance [6]. Conversely, contact-dependent mechanisms require cell-to-cell contact to deliver their toxic payload, allowing bacteria to kill a narrower range of microbial rivals that are within proximity [6]. These various types of short- and long-range weapons provide a competitive edge, arming distinct bacterial groups with the ability to target and kill microbial opponents.

Within the *Vibrionaceae* family, species such as *Vibrio cholerae* and *V. parahaemolyticus* often employ various antibacterial mechanisms to navigate and thrive within complex marine niches. Among these *Vibrio* species, the contact-dependent mechanism known as the type VI secretion system (T6SS) plays a key role in driving species fitness and survival across ecological niches [7]. Recently, several studies have characterized the functional role of T6SS core protein components, as well as some accessory proteins, expanding our understanding of the dynamics that govern T6SS assembly and regulation [8,9,10,11]. On a structural level, the T6SS can be described as a contractile nanomachine comprising several protein complexes that collectively resemble an inverted T4 bacteriophage tail [10]. When competitor cells eventually encounter one another, the T6SS functions by translocating toxic effector proteins from a toxin-producing cell into the membrane of a susceptible target cell [7]. These toxic effectors range from peptidoglycan hydrolases that target the cell wall to phospholipases that attack the cell membrane and nucleases that degrade target cell DNA and RNA [9]. This review highlights the broad arsenal of antibacterial weapons, ranging from diffusible toxins to contact-dependent systems that bacteria can deploy to engage in direct competition with prokaryotic opponents within their environment. Among these antibacterial weapons, we highlight the functional role of the T6SS in mediating intra- and interspecific bacterial competition across various ecological niches. We focus on key species within the *Vibrionaceae* family to examine the role of T6SS-mediated competition in shaping bacterial community structure and composition within marine environments and among host-associated communities. Additionally, we discuss the emerging roles of T6SS competition in influencing biofilm dynamics and horizontal gene transfer (HGT).

## 2. The Bacterial Arsenal of Antimicrobial Weapons

### 2.1. Diffusible Toxins

Bacteriocins are ubiquitous in the microbial world and are present in nearly all major groups of bacteria, as well as in some Archaea. They comprise a diverse group of diffusible multi-domain antimicrobial proteins, which are produced and released by an attacker cell [12]. These antimicrobial proteins are synthesized in the cytoplasm and can occasionally undergo post-translational modifications [13,14]. Bacteriocins are typically encoded within gene clusters that contain all the necessary gene products required for synthesis and export via cell lysis, as well as an immunity gene whose protein product confers the producing cell with protection against self-intoxication [12]. In contrast to broad-spectrum antibiotics, bacteriocins have a relatively narrow killing spectrum and are typically only toxic against bacteria that are closely related to the producing strain [14]. This narrow killing spectrum can be attributed to the specificity of bacteriocins for their target cell receptor.

There are various types of bacteriocins, and these are typically classified based on length and size, and on whether they contain modified residues. Class I bacteriocins, also referred to as lantibiotics, are small peptides (<5 kDa), comprising 19–35 amino acids that contain modified amino acid residues (lanthionine), and are commonly secreted by Gram-positive bacteria [15,16]. Class II bacteriocins, such as plantaricin EF, are small (<5 kDa) unmodified peptides produced by Gram-positive bacteria that can dimerize to form porin structures in cellular membranes [17,18]. Class II bacteriocins can be further classified into four subclasses (Class IIa-d) based on their distinct structural properties [16]. Microcins are small bacteriocins (<10 kDa) comprising 15–60 amino acids that are typically produced by members of the *Enterobacteriaceae* family [19,20,21]. Microcins can be further subdivided into Class I microcins if they contain post-translational modifications, or into Class II microcins if they remain unmodified [19].

In addition to these smaller bacteriocins, bacteria also produce several larger proteinaceous antimicrobials, such as colicin and colicin-like bacteriocins. Colicin proteins are large (40 to 80 kDa), multi-domain proteins that are typically produced by Enterobacteria and are released upon the lysis of the producer cell [22,23]. This self-sacrificing mechanism of release proves lethal for both the colicin-producing cell and neighboring bacterial cells whose extracellular receptors are recognized by these antimicrobial compounds [22]. Due to their double-edged nature, bacteriocins are typically only synthesized as a stress response mediated by the SOS regulon [23]. All colicins generally comprise three modular domains, each of which is involved in either binding to a receptor (the central domain), allowing translocation into target membranes (N-terminal domain), or carrying out toxin activity [23]. Thus, colicins function by first binding to their receptors located on the extracellular surface of target cells and are then transported across the outer membrane and into the periplasm via a Tol or TonB system [23]. These bacteriocins exert their lethal action in two ways: by mediating the formation of cellular membrane pores and by degrading target cell DNA, rRNA, and tRNA via their endonuclease activity.

### 2.2. Contact-Dependent Systems

In addition to the wide variety of freely diffusible bacteriocins, bacteria also employ competitive mechanisms that are activated only when in close proximity to potential competitors. As such, some bacteria can harbor complex toxin systems that allow them to sense and directly respond against competitors in a contact-dependent manner. These contact-dependent mechanisms typically possess membrane- or cell envelope-embedded functions that enable the translocation of toxins into target cells [24,25]. In contrast to diffusible bacteriocins, which are only released upon lysis of the producing cell, these contact-dependent systems enable bacteria to dispatch cytoplasmically synthesized toxic effectors into target cells without the need to self-sacrifice. These toxin-delivery nanomachines include the type IV secretion system, the contact-dependent inhibition (CDI) system (closely related to the type V secretion system), and the type VI secretion system [24,25,26].

The CDI system was first discovered in uropathogenic *Escherichia coli* and was the first toxin-delivery mechanism to be characterized as a contact-dependent antagonistic system [27]. This inhibition system is a specialized form of a two-partner secretion pathway, belonging to the type V secretion system family that is widely distributed among Proteobacteria [28,29]. The CDI system comprises two proteins, CdiA and CdiB, which encode for a secreted effector protein and outer membrane transport protein, respectively [27]. CdiA adopts an elongated β-helical structure with a C-terminal toxin domain that extends away from the cell surface [13,26]. This C-terminal domain mediates cellular surface recognition and toxic activity in susceptible bacteria. The CdiA effector protein is attached to CdiB, a β-barrel transport protein embedded in the outer membrane that facilitates the anchoring and delivery of CdiA into the periplasm of the target cell [2,30]. To prevent self-inhibition, the CDI system also contains an immunity protein, CdiI, which binds to the CdiA protein and neutralizes its toxic effect [29,30]. Stable contact with the susceptible target cell is required for binding of CdiA to a distinct outer membrane receptor found on the target cell surface [31,32]. Once contact is established, the CdiA protein undergoes self-cleavage leading to the release of the C-terminal toxin domain and its subsequent transport into the target cell. Once inside the target cell cytoplasm, cleaved CdiA toxins exhibit distinct nuclease activities that inhibit the growth of target cells [30,33]. Like bacteriocins, CDI^+^ bacterial cells are only able to target a narrow range of bacterial targets (such as close relatives) that possess the required membrane receptor on their cell surface [2,29,34].

The type IV secretion systems (T4SS) are multiprotein complexes that enable the direct transfer of DNA and effector proteins between bacterial cells via a contact-dependent mechanism [35]. The T4SS generally comprises a membrane-spanning channel and an extracellular pilus that allows for DNA and protein transfer in an ATP-dependent manner [36]. These nanomachines can be categorized into three subfamilies: (1) conjugation systems, (2) effector translocation systems, and (3) DNA uptake and release systems [35,37,38]. In both Gram-positive and Gram-negative bacteria, T4SSs mediate interspecies conjugation via pili-based interactions to transfer mobile DNA elements [35,37]. In certain Gram-negative bacteria, the T4SS has been shown to function as a translocation system for delivering toxic effectors into eukaryotic hosts [36,38,39]. T4SS also mediates the exchange of DNA with the extracellular milieu in a contact-independent manner by functioning either as a DNA-uptake system or as a DNA-release system [35]. In the last decade, studies have shown the T4SS to confer *Xanthomonas citri* with the ability to kill two different Proteobacterial species via the secretion of toxic effectors [40]. This recent finding suggests that the T4SS may play a larger role in mediating interbacterial antagonism than was previously believed. In addition to the T4SS, the type VI secretion is another contact-dependent mechanism that also plays a large role in mediating antagonistic behaviors among bacteria.

## 3. The Type VI Secretion System (T6SS)

### 3.1. General Background

The T6SS is one of many specialized protein secretion systems that bacteria have evolved to compete with one another. This secretion system is found in approximately 25% of all sequenced Gram-negative bacteria and is most predominant among proteobacterial genomes [41,42,43]. The firing of T6SS weapons is initiated upon cell-to-cell contact, which facilitates delivery of effector proteins into neighboring prokaryotic and eukaryotic target cells [7,9,10]. Since many of these effectors are potent toxins that target and disrupt cellular structures (e.g., cell wall and membrane), the T6SS-mediated translocation of these toxic effectors into a target cell often results in lethal cellular damage. The T6SS system and its associated genes were first described as playing a role in virulence in *R. leguminosarum* and *E. tarda*; however, the protein translocation activity of the T6SS was first characterized in *V. cholerae* [44,45,46,47,48]. Although initial interest in the T6SS centered around its role as a potential virulence factor utilized by pathogenic bacteria to mediate host colonization [46,48,49,50], more recent studies have cemented the role of the T6SS as a toxin delivery machine which mediates competitive interactions between bacterial neighbors [51,52,53,54]. Furthermore, T6SS-mediated antagonism has been shown to facilitate bacterial competition in both an inter- and intraspecific manner [55,56,57,58,59]. One important aspect of the T6SS is that it encodes cognate immunity genes, whose products neutralize the activity of T6SS toxins to hinder both self-intoxication and growth inhibition among isogenic cells [7,60,61]. Due to its antagonistic nature, the presence of T6SS weapons among diverse bacterial groups, such as the *Vibrionaceae*, may serve as a driving force in shaping the structure and composition of microbial communities across various marine ecosystems.

### 3.2. Assembly and Function of T6SS Proteins

While the T6SS is now widely recognized as a potent and aggressive antibacterial weapon, initial studies that focused on the function of structural T6SS proteins were crucial for our understanding of how this system is assembled and deployed for bacterial competition. These studies utilized cryo-electron microscopy and x-ray crystallography techniques to reveal that the T6SS apparatus is composed of 13 core protein subunits [9,62,63,64,65,66,67,68]. These core structural components are typically encoded by contiguous gene clusters that can average over 20 kb in length [49,69,70]. Core protein components are labeled with the suffix Tss [68,71], which stands for type six secretion, and are classified into three categories: contractile sheath-tube proteins (Hcp, VgrG, TssA, TssB, TssC, TssH), baseplate-associated proteins (TssE, TssF, TssG, TssK), and membrane-associated proteins (TssL, TssM, TssJ) [9]. Additionally, several accessory proteins are also commonly found within T6SS gene clusters. While a majority of these accessory proteins have no inferred functional role, a few appear to play a vital part in allowing the proper functioning of the apparatus [72,73]. Core protein subunits of the T6SS assemble into three distinct substructures: (1) a sheath-tube structure that resembles a syringe and contracts to translocate effector proteins across bacterial membranes; (2) a baseplate complex that acts as a platform for the sheath-tube structure; and (3) a membrane-associated complex that anchors the sheath-tube structure to the inner bacterial membrane [9] (Figure 1). Together, these T6SS substructures comprise a transmembrane protein complex that resembles an inverted T4 bacteriophage tail.

#### 3.2.1. Contractile Sheath-Tube Proteins

After the initial finding of a T6SS in *V. cholerae*, two proteins belonging to the haemolysin coregulated protein (Hcp) and the valine-glycine repeat G (VgrG) protein families were the first functional proteins to be identified in the T6SS. The crystal structure of Hcp and its homologs revealed that this protein polymerizes into stacked hexameric protein rings that form a tubular structure upon T6SS assembly [49,74]. At the tip of the Hcp tail tube, VgrG proteins assemble into a trimeric complex that forms a spike-like structure resembling an arrowhead [67]. This Hcp-VgrG complex is structurally similar to the tail-spike of T4 phages that are responsible for puncturing the host cell wall [67]. In a similar fashion, the Hcp-VgrG complex acts as a poison-tipped arrow that penetrates the cellular membrane of target cells to deliver toxic effectors. Two other T6SS components, TssB and TssC, assemble into a hollow helical structure with a cog wheel-like appearance [75,76]. TssB/C tubules form a contractile sheath that encases the inner T6SS tail-spike. The TssB/C sheath functions similarly to the T4 bacteriophage tail-sheath by translocating the inner Hcp-VgrG tubule across the cellular membrane via sheath contraction [75,77]. Upon the initial puncturing of a target, VgrG and Hcp proteins are released into the target cell cytoplasm, while some are also secreted into the extracellular milieu. A fluorescent TssB-GFP fusion protein was employed to visualize tail-sheath formation in *V. cholerae*, revealing that sheath proteins oscillate between cycles of assembly, extension, contraction, and disassembly [78]. Located at the end of the sheath, the TssA cap protein recruits the baseplate complex and coordinates the priming and polymerization of the Hcp tube and the TssB/C sheath to ensure proper sheath elongation within the cytoplasm [79,80]. Following T6SS firing, TssH (an ATPase) is responsible for disassembling the contracted T6SS sheath via ATP hydrolysis [10,81]. Sheath depolymerization enables the reuse and recycling of TssB/C monomers for future attacks.

#### 3.2.2. Baseplate-Associated Proteins

The T6SS tail-sheath structure has been shown to associate with the inner bacterial membrane via a large baseplate complex that serves a similar function to the baseplate of T4 bacteriophage. This baseplate complex, composed of TssE, TssF, and TssG proteins, serves as a platform that anchors sheath proteins to the membrane complex to ensure proper sheath assembly and contraction [78]. Within this baseplate complex, TssE, TssF, and TssG proteins work together to form a structural scaffold to promote the stability of the baseplate complex, which in turn ensures the proper attachment and polymerization of sheath proteins to prevent premature sheath contraction [62]. Following sheath assembly, TssE is also responsible for triggering sheath contraction upon receiving an activation signal [67]. This activation signal is mediated by TssK, which facilitates the translocation of environmental signals to the baseplate complex and serves as a link connecting the baseplate complex to the membrane complex [78,82].

#### 3.2.3. Membrane-Associated Proteins

In addition to the baseplate complex, a membrane-associated complex related to the T4SS family also helps to anchor the contractile tail-sheath structure to the membrane. This T4SS-like anchoring complex consists of an outer membrane lipoprotein, TssJ, and two inner membrane components, TssL and TssM [83]. TssL localizes to the cytoplasm and is anchored to the inner membrane via transmembrane domains found on its C-terminus [84,85,86]. TssL is structurally related to the DotU protein of the T4SS, which forms a complex with IcmF that helps stabilize the core components of the T4SS apparatus [87,88]. The IcmF homolog in the T6SS is TssM, which predominantly localizes within the periplasm and is anchored to both the inner and outer membrane [89]. TssM spans across the entire cell envelope and acts as a connector between the TssL inner membrane protein and the TssJ outer membrane protein [89]. TssJ is a lipoprotein that is anchored to the outer membrane via its N-terminal domain and contains a variable loop that interacts explicitly with TssM [89]. Altogether, TssL/TssM/TssJ form a membrane anchoring complex analogous to those observed in the T4SS.

#### 3.2.4. Effector Proteins

T6SS effectors come in various forms and display enzymatic activities that range from muramidase and lipases, which target the bacterial cell wall and membrane, to peptidases and nucleases, which target bacterial proteins and DNA. These effectors can be secreted as specialized C-terminal extensions on VgrG spike proteins or as separate cargo [60,90,91]. All strains of *V. cholerae* contain multiple VgrG alleles that are encoded in either the large or auxiliary T6SS clusters [57]. In some *V. cholerae* isolates, the VrgG1 protein located in auxiliary cluster 1 (Aux1) contains a C-terminal domain that displays anti-eukaryotic activity by cross-linking actin monomers, resulting in the disruption of the cytoskeleton [45,86]. Found in the large T6SS cluster, VgrG3 is another effector protein that harbors a C-terminal domain with antibacterial activity [60]. VgrG3 exhibits lytic activity and contains a predicted muramidase fold that can degrade the peptidoglycan (PG) cell wall [60]. Interestingly, nonidentical VgrG proteins can assemble to form a heterotrimeric spike complex, which in turn allows the T6SS to deliver more than one effector domain within a single payload [45].

Besides VgrG proteins, T6SS auxiliary clusters also encode cargo effectors that are loaded onto the VgrG spike complex via the aid of chaperone proteins. One of the first T6SS cargo effectors to be biochemically characterized was the Tse1 (type VI secretion exported 1) effector protein of *Pseudomonas aeruginosa*, which exhibits PG degrading activity [53,92]. In *V. cholerae*, TseH is a similar T6SS cargo effector that displays cell-wall targeting activity and is found within auxiliary cluster 3 (Aux3) [93]. TseH is an endopeptidase that contains an amidase domain which breaks down the PG cell wall and was found to be toxic against *Aeromonas* and *Edwardsiella* species [93]. Like the cell wall, the cell membrane is also an essential component of bacterial cells and, therefore, serves as another crucial target for T6SS effectors. In *V. cholerae*, TseL is another cargo effector that is encoded within Aux1 and is loaded and secreted via the VgrG1 protein that is also encoded within Aux1 [94,95,96]. TseL is a lipase protein that belongs to the Tle2 family of lipases that display membrane-targeting activity against both prokaryotic and eukaryotic cells [95]. In addition to phospholipases, there are also pore-forming proteins that target bacterial membranes. For example, the first pore-forming T6SS cargo effector to be described was VasX from *V. cholerae*, which exhibits a similar phenotype to that of pore-forming colicins [97,98]. The formation of these pores disrupts the cell membrane potential and compromises membrane integrity, leading to cellular leakage that results in cell death.

The diversity of T6SS effector proteins and the cellular components they target—from the cell wall to the membrane—demonstrates the potency of this system in combating bacterial rivals. However, it is crucial to note that the effects of these toxic proteins can be neutralized if the targeted cell carries a cognate immunity protein that inhibits the activity of the T6SS effector. Thus, the presence or absence of immunity proteins against the array of T6SS effectors deployed by an attacking cell ultimately determines whether a bacterial cell survives or succumbs to cell lysis. Furthermore, bacteria can evolve ingenious defense mechanisms to circumvent the lethal effects of these toxins.

## 4. T6SS in Vibrio: Impact on Marine Free-Living and Host-Associated Microbial Communities

By serving as a toxin-delivery machine, the T6SS confers *Vibrionaceae* species with a competitive edge to combat bacterial opponents, allowing these species to assert their dominance over an ecological niche across various marine habitats. Within bacterial communities, clonal or sister cells can usually coexist in the same niche because they are compatible—they encode cognate immunity genes capable of neutralizing T6SS attacks brought on by sister cells [99,100]. However, if neighboring cells do not encode cognate immunity proteins against the T6SS effectors of the attacking cell, they will be subject to cell lysis [7,92,101]. This incompatibility between bacterial cells is typically observed between interspecies T6SS-mediated interactions. The next sections of this review will describe T6SS clusters and their functions within key *Vibrio* species, highlighting how these protein secretion mechanisms mediate intra- and interspecific competition within free-living marine microbial communities, host-associated communities, and in other specialized ecological niches. To facilitate comparisons, Table 1 provides an overview of the Vibrio species discussed below and the diversity and function of their T6SSs.

### 4.1. T6SS of V. cholerae in Marine Microbial Communities

*V. cholerae*, the causative agent of cholera disease in humans, was one of the first model organisms in which the antibacterial function of the T6SS was initially studied. In most *V. cholerae* strains, the T6SS is encoded by three distinct gene clusters: the large cluster and auxiliary clusters 1 and 2 [91,102]. The large cluster encodes the core structural components (e.g., *hcp*, *vrgG*, *tssB/C*, *tssM/L*) of the T6SS apparatus, allowing *V. cholerae* to assemble and fire this weapon [102]. Auxiliary cluster 1 encodes additional *vgrG* and *hcp* isoforms, as well as specialized effector-immunity pairs (e.g., *vasX*, which targets the membrane) that increase the breadth of bacterial competitors that can be targeted [57,103]. Auxiliary cluster 2 encodes regulatory proteins that control T6SS activation in response to environmental cues, to ensure that T6SS harpoons are only active under favorable conditions, allowing *V. cholerae* to maintain its competitive edge without unnecessary energy expenditure [55]. Many environmental strains also contain a third, fourth, or fifth auxiliary cluster that encodes additional T6SS effector and accessory proteins that enhance the adaptability of this secretion system across various environmental and host environments [55,103]. The ability to acquire modified or specialized T6SS effector genes through auxiliary clusters contributes to the diversity of T6SS effector-immunity module sets that are found among *V. cholerae* strains.

To date, most sequenced pathogenic *V. cholerae* strains harbor compatible T6SS effector-immunity module sets. Additionally, pathogenic strains of *V. cholerae* often harbor T6SSs that are inactive or tightly regulated by environmental signals (e.g., nutrient availability, salinity, and bile salts) or quorum sensing [55,91,104]. This hinders pathogenic *V. cholerae* strains from engaging in aggressive bacterial killing that could trigger host immune responses, prioritizing energy allocation towards essential virulence functions required for host colonization. By contrast, nonpathogenic environmental strains display highly variable effector-immunity pairs throughout T6SS auxiliary clusters [57,58]. These environmental strains are typically regarded as ‘incompatible’, since a high diversity of T6SS effector repertoires decreases the likelihood of strains coexisting in the same niche. Such diversity in effector-immunity module sets contributes to both intra-genus and intra-species competition and clonal segregation among *V. cholerae* strains across ecological niches [55,57,105,106] (Figure 2(1)). Unlike pathogenic strains, environmental *V. cholerae* strains possess T6SS weapons that are constitutively armed and active [106,107]. Among a subset of environmental strains, this constitutive T6SS expression allowed smooth *V. cholerae* strains to successfully eliminate *E. coli* and other environmental bacteria, in addition to the social amoeba *Dictyostelium discoideum* [58]. Furthermore, these environmental *V. cholerae* strains were also able to outcompete each other in a T6SS-dependent manner [58]. The constant T6SS-ON state observed among environmental *V. cholerae* strains highlights an ecological strategy employed by *Vibrio* species to dominate within aquatic habitats, where resources and niche space are limited, via T6SS-mediated antagonism. This competitive strategy is critical for ensuring a winning outcome among increasingly unstable marine ecosystems.

### 4.2. T6SS of Vibrio parahaemolyticus in Marine Microbial Communities

Beyond *V. cholerae*, *Vibrio parahaemolyticus*—the causative agent of gastroenteritis in humans—is another marine bacterium that employs T6SSs to eliminate microbial competitors and enhance it environmental fitness. This emerging pathogen harbors multiple, distinct T6SS that play pivotal roles in both pathogenicity and ecological adaptability [108]. Comprehensive genomic analyses of the *V. parahaemolyticus* pan-genome have identified four distinct T6SS gene clusters [108]. Two (T6SS1 and T6SS2) appear to be ancient and widespread, while the other two (T6SS3 and T6SS4) are rare in nature and likely acquired via HGT. T6SS1, predominantly found among pathogenic isolates, is regulated by both quorum sensing and surface sensing mechanisms and is activated under warm, marine-like conditions that bestow a competitive advantage against other microbial species in the marine environment [108,109,110]. Under simulated seawater conditions at 30 °C, T6SS1 exhibits potent antibacterial activity against *V. cholerae* and other Proteobacteria [110]. T6SS2, by contrast, is activated under low-salinity conditions and is broadly conserved among *V. parahaemolyticus* isolates [108,110]. Although T6SS2 does not appear to mediate direct bacterial competition, it plays a crucial role in environmental adaptability and host interactions by promoting the cellular adhesion of *V. parahaemolyticus* to HeLa Cells and inducing autophagy in macrophages [111]. While host-related functions of T6SS2 have not yet been extensively studied in marine hosts, current evidence indicates that *V. parahaemolyticus* adopts a dual strategy of utilizing T6SS1 to eliminate microbial competitors while simultaneously utilizing T6SS2 to enhance host cell adhesion, thereby facilitating colonization and persistence within the host niche. Unlike the other systems, T6SS3 and T6SS4 are relatively rare among *V. parahaemolyticus* isolates (present in >2% of analyzed genomes), and their limited distribution suggests they confer an advantage only under select environmental conditions, but their exact function is not yet understood [108]. Further research is needed to uncover their role in the ecology and pathogenicity of *V. parahaemolyticus*. In addition, genomic analyses identified diverse T6SS auxiliary modules containing known or predicted putative effectors, as well as novel T6SS effectors that may contribute to the environmental fitness of *V. parahaemolyticus* across diverse aquatic milieu [108].

### 4.3. T6SS of Vibrio coralliilyticus in Coral Symbiosis

While many *Vibrio* species exist as planktonic cells within the water column, some species are opportunistic pathogens that alternate between free-living and host-associated states. During this phase transition, T6SS deployment enables *Vibrio* species to establish and dominate a host niche. Within the complex environment of coral reefs, microbial symbionts play a crucial role in nutrient acquisition and cycling and safeguarding against pathogens, which is essential for maintaining coral health [112,113,114]. Thus, the deployment of T6SS weapons within these fragile ecosystems by opportunistic *Vibrio* species can have profound implications for both coral microbiome composition and disease dynamics. Recent studies on the opportunistic coral pathogen *Vibrio coralliilyticus* have revealed that this opportunistic bacterium harbors two distinct type VI secretion systems (T6SS), both of which are upregulated under elevated temperatures [115,116]. The first system (T6SS1) encodes a suite of antibacterial effectors that facilitate the ability of *V. coralliilyticus* to directly target and kill commensal bacteria within the coral microbiome, as illustrated in Figure 2(2) [115,117]. Alternatively, the second system (T6SS2) delivers a toxic payload of novel anti-eukaryotic effectors into host cells [115,116,118]. These findings suggest that during periodic marine heatwaves driven by climate change, *V. coralliilyticus* can deploy both of its T6SS to either directly damage coral host tissues or disrupt the coral microbiome by eliminating beneficial symbionts. This dual targeting within the coral symbiosis could lead to devastating coral disease outbreaks and the subsequent degradation of coral reefs across the globe.

Beyond corals, *V. coralliilyticus* can also utilize its T6SS-delivered toxins against a range of other marine organisms, including crustaceans. In the aquatic host model, *Artemia salina*, T6SS2 mediates the secretion of novel anti-eukaryotic toxic effectors that contribute to larval (*Artemia* nauplii) mortality [119]. Additionally, this study also demonstrated that the T6SS1 of *V. coralliilyticus* exhibited antibacterial killing activity against the target prey *V. natriegens* and two other *Vibrio* species associated with infections among aquaculture, *V. alginolyticus* and *V. campbellii* [119]. Collectively, these findings illustrate the multifaceted role of the T6SSs of *V. coralliilyticus* in facilitating both the elimination of interbacterial competitors and mediating host virulence to colonize distinct animal niches across marine ecosystems. This functional adaptability of T6SS weapons targets a broader evolutionary trajectory in marine bacteria, in which T6SSs have been adapted not only to facilitate pathogenicity but also to foster beneficial partnerships with host organisms. While the T6SSs of opportunistic pathogens like *V. coralliilyticus* inflict direct damage on host tissues and disturb the microbiome, other marine species have adapted this nanoweapon to aid in establishing mutualistic associations. In recent years, one species in particular, *V. fischeri*, has garnered significant attention for its use of T6SS weaponry to aid in the successful colonization of its squid host [59].

### 4.4. T6SS of Vibrio fischeri in Squid Symbiosis

The symbiosis between sepiolid squids (Cephalopoda: Sepiolidae) and the bioluminescent bacterium *V. fischeri* represents a valuable model system for studying microbial interactions and the impact of T6SS competition in a symbiotic system [120]. In this mutualism, *V. fischeri* symbionts provide the host with protective camouflage via silhouette reduction against predators while the squid provides the symbionts with a safe dwelling against environmental stressors [121,122,123]. Squid hatchlings are born aposymbiotic and acquire their bacterial partners from the environment—these colonize a specialized bilobed structure known as the light organ (LO) located in the squid mantle [124,125]. Each side of the LO features three independent crypt spaces that serve as potential colonization sites [126]. Once in the LO, *V. fischeri* symbionts form biofilms and multiply to a high cell density, triggering bioluminescence that the nocturnal squid can use to counterilluminate their moon-lit shadow in the water column [123]. After the evening at dawn, the host expels ~95% of the symbiont population, while the remaining 5% repopulate the LO to repeat this diel cycle [122]. Earlier studies on the squid-*Vibrio* mutualism revealed that only a few strains (at most six) typically colonize the LO of wild-caught animals, even though many strains of *V. fischeri* are naturally present in seawater [127]. Since the LO represents a haven for bacterial replication (e.g., increased fitness), researchers speculate that *V. fischeri* symbionts might engage in direct competition to occupy this niche.

Evidence of interference competition within the LO first arose from squid colonization studies in which aposymbiotic animals were exposed to two distinct *V. fischeri* strains, FQ-A001 and ES114. Results revealed that while the LO of these animals could be colonized by both strains, each of the crypt spaces was monospecific and colonized by either one of the strains but not both [128]. Subsequent studies utilized mutants of FQA001 that were impaired for T6SS activity to show that the strain incompatibility observed in the LO crypts was dependent on the T6SS of FQ-A001 [59,129]. Whole-genome sequencing later revealed that a subset of all sequenced *V. fischeri* strains, including FQ-A001, encode a T6SS cluster on chromosome II known as T6SS2, while strains like ES114 do not encode this gene cluster [130]. *V. fischeri* strains encoding this gene cluster (e.g., FQ-A001) are known as ‘lethal’ strains due to their ability to directly kill or eliminate ES114 in a T6SS2-dependent manner in culture-based assays and in vivo [59]. Together these results indicate that the T6SS2 cluster of *V. fischeri* is strain-specific and facilitates intraspecific competition among mutualistic symbionts to allow ‘lethal’ strains to outcompete and kill susceptible *V. fischeri* strains within the LO crypts during initial host colonization (Figure 3).

Before entering the LO, the host colonization process also involves environmental bacteria forming large aggregates on the nascent LO surface, which gradually become enriched with *V. fischeri* cells over time [131]. During this aggregation stage, *V. fischeri* symbionts are forced into contact with other environmental bacteria and must compete with these bacterial opponents to dominate at the surface pores and gain access to the LO ducts that lead into the deeper crypt spaces [122,131]. Results from additional in vitro studies have also shown that lethal *V. fischeri* strains are able to eliminate susceptible strains when grown in a neutral/acidic (pH 6.5 and 7.5) hydrogel medium akin to the host mucus that is secreted from the LO pores [132,133]. Interestingly, lethal *V. fischeri* strains were not able to outcompete their target prey when cultured in a basic (pH 8.2) hydrogel medium, which mimics the pH conditions of seawater [133]. These findings suggest that as lethal *V. fischeri* strains transition from their free-living state in seawater to their symbiotic state within *Euprymna*, host-derived mucus activates T6SS2, while the neutral/acidic conditions of the LO microenvironment promote direct cell-to-cell contact. Together, these host-associated environmental cues enable *V. fischeri* to eliminate both environmental rivals at the surface pores and other strains of *V. fischeri* within its natural host. Thus, the T6SS2 of *V. fischeri* facilitates the ability of this beneficial symbiont to engage in both inter- and intraspecific competition during early host colonization to successfully establish and dominate within the LO of their squid host. These findings highlight the evolutionary specialization of T6SS weapons for specific ecological niches, where their function may be restricted for activation only under host-associated conditions, such as in the presence of host-derived mucus and acidic microenvironments. In this scenario, bacterial cells have a higher possibility of coming into direct cell-to-cell contact with potential competitors. In open marine environments, the lowered frequency of direct cell contact decreases the likelihood that cells will engage in direct competition, thereby rendering the deployment of this system both futile and energetically wasteful. By restricting T6SS activity to the host environment, bacterial cells can limit the metabolic burden of synthesizing this large apparatus and ensure its deployment is reserved for critical stages, such as during early host colonization. This host-dependent activation sheds light on a broader evolutionary strategy employed by some *Vibrios*, where T6SS weapons no longer function as a constitutively active offensive combat system, but rather as a fine-tuned instrument that responds to signals from the host microenvironment, ensuring successful colonization and persistence within its symbiotic host.

All sequenced *V. fischeri* strains also harbor an additional T6SS cluster on chromosome I, referred to as T6SS1, which encodes for 11 of the T6SS structural components except for the *hcp* and *vgrG* genes [59]. Disruption of T6SS1, however, had no impact on the T6SS antibacterial activity of lethal *V. fischeri* strains [59]. While these findings suggest that only T6SS2 contributes to the T6SS activity and lethal phenotype of strains like FQ-A1001, it is possible that the T6SS1 cluster is only activated under specific environmental conditions that have not yet been identified or that it mediates alternative functions in *V. fischeri*. Since most studies of the T6SS in *V. fischeri* have focused on the competitive advantage of this weapon, further research needs to be conducted to determine whether *V. fischeri* may utilize T6SS1 in an alternative mechanism.

Interestingly, although T6SS2 provides *V. fischeri* strains with a distinct competitive advantage during early host colonization, this cluster is strain-specific and is only found in about 70% of all sequenced symbiotic *V. fischeri* genomes [134]. This suggests that although deploying T6SS2 may be advantageous under a host-specific context, there may be selective pressures that limit its distribution across all strains. Some of these potential constraints include the inadvertent triggering of host immune responses via excessive antibacterial killing, the potential for cooperation among symbionts to outweigh competitive dynamics in support of their functional contribution to the host, and the possibility that over time T6SS weapons may incur a metabolic cost which negatively impacts symbiont fitness. For instance, a recent study demonstrated that quorum-sensing molecules released by a susceptible *V. fischeri* strain (ES114) inhibit T6SS-mediated competition, allowing the ES114 competitor to coexist with the lethal *V. fischeri* strain within the same LO crypt [135]. This study provides initial evidence that certain mechanisms may modulate cooperation between lethal strains and their susceptible competitors, facilitating microbial coexistence and the fulfillment of their symbiotic roles within the host environment. Additional findings have also shown that in lethal *V. fischeri*, T6SS2 incurs a minor fitness cost—reduced cell density at the stationary phase under host-like conditions, although this effect was not observed in seawater-like conditions [136]. Over time, the cumulative fitness cost of activating this system could become increasingly detrimental. Furthermore, an in silico study demonstrated that if T6SS deployment imposes a high energetic burden, it could create a tipping point in lethal *V. fischeri* strains where expression of this weapon would become self-defeating, since associated costs would outweigh any competitive advantage [72,136]. In this scenario, a strain lacking the T6SS could potentially outcompete a lethal strain by allocating resources to promote faster growth and expansion to overtake the lethal competitor.

Additionally, while T6SS competition may provide *V. fischeri* strains with an advantage during initial host colonization, once symbionts have successfully established within the host niche, the continued use of T6SS weapons could be detrimental to symbiont fitness due to their metabolic burden. This, in turn, may place a selective pressure on lethal *V. fischeri* strains to decommission their T6SS weapons, potentially through the loss of essential T6SS genes or even the entire gene cluster [136]. This scenario is further supported by the fact that not all strains of *V. fischeri* isolated from the wild encode the T6SS2 cluster, suggesting that the evolutionary trajectory of this weapon may not be universally advantageous across all life stages of this symbiont. While it is clear that T6SS2 provides a competitive edge to lethal *V. fischeri* strains during the early stages of host colonization, the use of this weapon does not guarantee symbiotic success. Similarly, encoding a potent T6SS arsenal does not equate to whether the most lethal *V. fischeri* strain will dominate within the host niche, as certain scenarios may favor strains that lack T6SS weapons. Altogether, these findings highlight the double-edged nature of T6SS weapons and emphasize how bacterial species or strains that encode for this system must strike a balance between competitor killing and the potential metabolic burden of their weapons.

## 5. Emerging Role of T6SS in Vibrio: Shaping Biofilm Dynamics

Like *V. fischeri*, many *Vibrio* species can also form biofilms on biotic and abiotic surfaces in marine habitats [137,138]. As planktonic cells transition from a free-living to a biofilm-associated state, the T6SS helps facilitate the structuring of microbial communities within the biofilm. Bacterial cells are forced into proximity within these biofilms, facilitating the cell-to-cell contact required for T6SS deployment to kill target cells [139,140]. As illustrated in Figure 2(3), these conditions allow T6SS active strains to establish dominance within a biofilm via the selective killing of competing strains, which in turn alters the community composition of the biofilm [141,142]. In mixed-species biofilms, *V. cholerae* employs T6SS weapons to eliminate various bacterial isolates belonging to the γ-Proteobacteria phylum, effectively reducing its competition for nutrients and space.

Besides community structuring, the T6SS also contributes to biofilm maturation by regulating the production of extracellular polymeric substances (EPS), which form a protective matrix that encases cells within the biofilm. This EPS matrix is essential for maintaining the structural integrity of the biofilm, and promotes its maturation and resilience, especially when exposed to environmental stressors (e.g., salinity fluctuations and antimicrobial agents) [143,144]. By upregulating both EPS production and T6SS activity, the dual action of VxrB allows *V. cholerae* to dominate within biofilm communities, enhancing its ability to form monospecies biofilms and resist displacement [145,146]. In *V. parahaemolyticus*, TssL2, which is a key component of T6SS2, promotes cellular attachment and aggregation via the regulation of genes related to motility and biofilm formation [147]. By modulating the initial adhesion of *V. parahaemolyticus*, T6SS2 enhances the ability of this bacterium to colonize and form biofilms on surfaces like shellfish, sediments, and marine equipment. IcmF2, another component of the T6SS2 of *V. parahaemolyticus*, has also been shown to promote the production of essential components of the biofilm matrix [148]. The dual role of the T6SS in shaping biofilm structure (via direct competitor elimination) and stability (via EPS production) promotes the formation and long-term persistence of *Vibrio* biofilms across various surfaces in the marine environment.

## 6. Emerging Role of T6SS in Vibrio: Facilitating Gene Acquisition via HGT

Among bacterial communities, the T6SS facilitates the elimination of microbial competitors by injecting toxins into susceptible cells, which inevitably leads to cell death or lysis of the target cell. The rupturing of these target cells results in the subsequent release of cellular components like DNA into the extracellular environment [92,149]. Consequently, T6SS-mediated killing generates a pool of environmental DNA that can be seized by neighboring bacterial cells through HGT mechanisms (Figure 2(4)). In some *Vibrio* species such as *V. cholerae* and *V. parahaemolyticus*, the presence of chitin serves as an environmental cue that induces the uptake of DNA via natural transformation pathways [150,151,152,153]. Chitin is a naturally occurring polysaccharide polymer that is prevalent within marine ecosystems and is primarily found in the exoskeletons of crustacea and other marine invertebrates [154,155]. Chitin polymers function by inducing the expression of the competence regulator, TfoX, which in turn activates the expression of DNA uptake machinery. In *V. cholerae*, studies have demonstrated that in the presence of chitin, TfoX activation coupled with T6SS-mediated killing of susceptible cells, releases DNA into the surrounding environment that can be actively acquired via HGT [149,156]. Within biofilms, where microbial communities are densely packed, the effects of T6SS-mediated killing can be particularly potent. In these tight-knit environments, released DNA can be readily seized by competent members within the biofilm and used as a nutrient source. Alternatively, the T6SS liberated DNA can also be recombined into the genome of recipient cells to promote the acquisition of novel virulence factors, antibiotic resistance genes, and traits that enhance the ecological fitness of a bacterial species. This process highlights the diverse functions of the T6SS, not only in driving intra- and interbacterial competition but also in promoting novel gene acquisition to enhance the diversification and adaptability of *Vibrionaceae* species across marine ecosystems.

## 7. Concluding Remarks

Bacterial competition plays a pivotal role in shaping the composition and structure of microbial communities across marine ecosystems. To secure access to limited resources (e.g., nutrients and space), bacteria can deploy a diverse arsenal of antibacterial weapons to warrant their dominance within an ecological niche. Among these, the T6SS stands out as a specialized contact-dependent nano-weapon capable of delivering a broad array of toxic effectors. By functioning as a toxin-delivery machine, the T6SS enables members of the *Vibrionaceae* family to engage in direct competition and effectively eliminate intra- and interspecific rivals. In marine-associated microbial communities, species like *V. cholerae* and *V. parahaemolyticus* are capable of outcompeting neighboring bacteria in a T6SS-dependent manner to maintain their dominance over a niche. Within host-associated microbiomes, like those found in coral symbiosis, the T6SS allows *V. coralliitycus* to displace resident members to facilitate host colonization and the onset of pathogenesis. More recently, T6SS-mediated competition has been shown to aid in establishing beneficial interactions. For example, in the mutualistic squid-*Vibrio* symbiosis, the T6SS of *V. fischeri* enables the elimination of symbiotic competitors to establish a stable relationship with its cephalopod host. Additionally, emerging evidence suggests that T6SS activity modulates biofilm composition and dynamics, potentially promoting HGT by facilitating the uptake and integration of extracellular DNA.

While recent research has significantly expanded our understanding of T6SS function in both the open marine environment and within distinct marine hosts, several key questions remain. Future research should aim to characterize the full repertoire of T6SS effectors and their cognate immunity proteins, as well as clarify the role of accessory proteins encoded in the diverse T6SS auxiliary clusters found across bacterial taxa. Moreover, there is a growing need to investigate how T6SS-mediated competition influences microbial community composition across a broader range of host-associated systems, particularly in underexplored aquatic hosts. Such investigations will be pivotal in uncovering the ecological impact and context-dependent activity of the T6SS across diverse marine environments. Although a few studies have explored the fitness costs associated with maintaining and deploying T6SS machinery, these efforts have largely been confined to shorter timescales. To address this limitation, future research should employ ecologically relevant model systems and long-term experimental evolution approaches to assess how T6SS influences bacterial fitness, adaptation, and stability within microbial communities. Together, these efforts will not only deepen our understanding of T6SS-mediated competition across marine systems but will also shed light on the broader ecological and evolutionary processes that govern microbial community dynamics in the ocean.

## Figures and Tables

**Figure 1 microorganisms-13-01370-f001:**
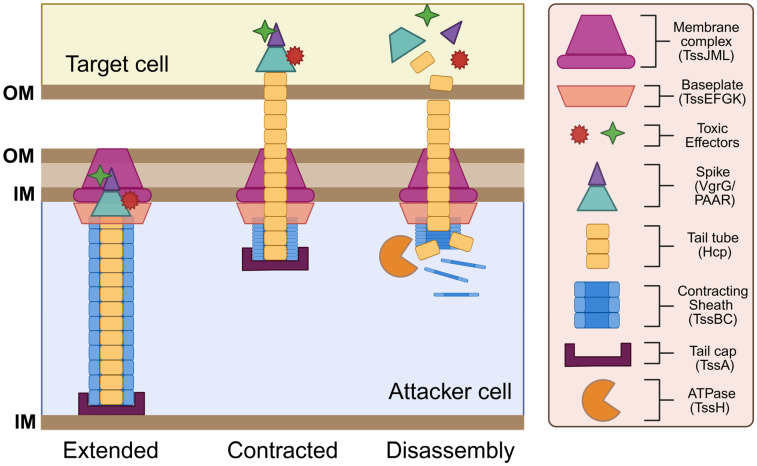
Structure and function of the T6SS. The T6SS apparatus comprises a needle-like tube with a poison-carrying tip that is encased by an outer sheath, which is anchored to the cell membrane by the baseplate and membrane complex. Components assemble in an extended state. Upon contact with a target cell, sheath contraction (firing) plunges the needle into the target cell. The tip of the needle disassembles to deliver the toxic effectors, while structural components in the attacker cell are recycled by the ATPase. This figure was created using Biorender.com.

**Figure 2 microorganisms-13-01370-f002:**
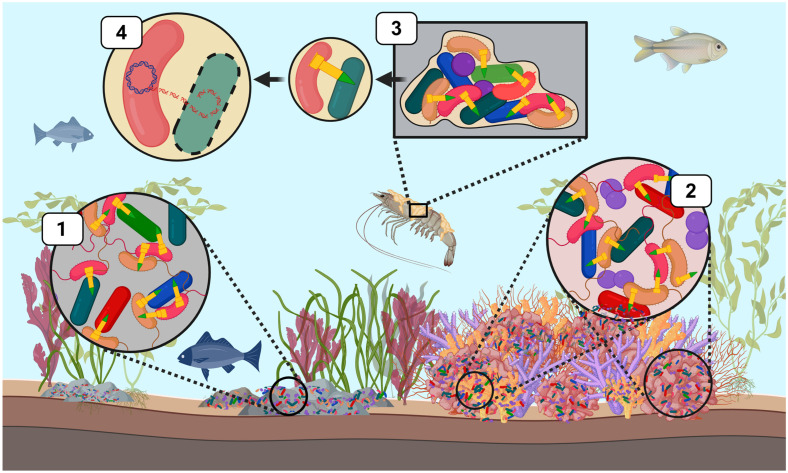
T6SS-mediated bacterial competition by *Vibrionaceae* across diverse marine ecological niches. T6SS weapons (yellow) have a central role in shaping marine microbial communities by mediating intra- and interspecific competition. This schematic illustrates ecological contexts in which T6SS facilitates the elimination of microbial competitors or provides alternative functions. (**1**) *V. cholera* (pink) and *V. parahaemolyticus* can deploy T6SS weapons to eliminate intra- and interspecific competitors in marine-associated microbial communities, such as those found on marine sediments and rocks. (**2**) *V. coralliilyticus* utilizes multiple T6SS clusters to directly target members of the coral microbiome and can also secrete anti-eukaryotic effectors that damage coral host tissue. (**3**) *Vibrio* species such as *V. cholerae* and *V. parahaemolyticus* form biofilms on abiotic and biotic surfaces (e.g., crustacean carapaces). These bacterial species can deploy T6SSs to target microbial members and alter the composition and spatial structure of species within the biofilm, where subsequent lysis of target cells facilitates T6SS-mediated HGT (**4**). This figure was created using Biorender.com.

**Figure 3 microorganisms-13-01370-f003:**
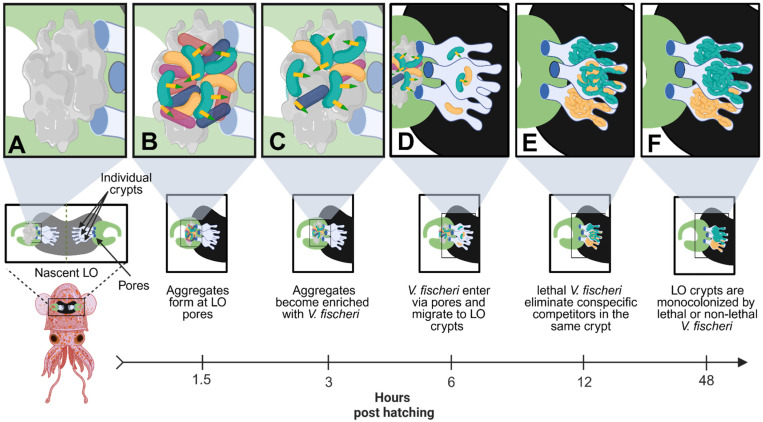
Lethal *V. fischeri* use T6SS2 to eliminate competitors during early host colonization. (**A**) Upon hatching, juvenile squids are aposymbiotic and begin secreting mucus at the surface pores of the nascent LO to attract bacteria from the surrounding environment. (**B**) Environmental bacteria, including multiple strains of *V. fischeri*, begin forming aggregates at the surface pores of the nascent LO. The host mucus activates the T6SS2 antibacterial weapons of lethal *V. fischeri* (green) cells. (**C**) Lethal *V. fischeri* use T6SS2 to eliminate environmental bacteria at the surface pores of the LO, which gradually enriches the aggregates with *V. fischeri* cells over several hours. (**D**) *V. fischeri* cells enter the nascent LO through the surface pores and migrate into one of three individual crypt spaces, found on each side of the bilobed LO. Multiple cells can enter each LO crypt, including both lethal (green) and non-lethal (yellow) *V. fischeri* cells, but typically only 1–2 cells will enter each crypt. (**E**) Once in the LO crypts, *V. fischeri* will begin to multiply until they reach a quorum, which turns on bioluminescence. If a lethal and non-lethal strain colonizes the same LO crypt, the lethal strain will begin to eliminate the non-lethal strain via a T6SS2-dependent manner. (**F**) After 48 h post-hatching, the lethal strain will outcompete the non-lethal strain, resulting in monocolonized crypts that contain either lethal or non-lethal *V. fischeri.* This figure was created using Biorender.com.

**Table 1 microorganisms-13-01370-t001:** Comparative overview of *Vibrio* species that encode T6SSs diversity, regulation, environmental cues, and ecological roles across marine niches.

Species	Number of Clusters	Type	Diversity Across Strains	Regulation	Activation (Environmental Cues)	Ecological Function in Marine Niches
*V. cholerae*	1 main + 2 auxiliary (up to 5 total)	Antibacterial and anti-eukaryotic	Conserved effectors among pathogens; high effector diversity among environmental strains	Tight in pathogens; constitutive in environmental strains	Bile salts, salinity, nutrient availability	Facilitates inter- and intrabacterial competition; Promotes dominance of environmental strains in seawater
*V. parahaemolyticus*	Four (T6SS1–T6SS4)	T6SS1: antibacterial. T6SS2: anti-eukaryotic; T6SS3/4: unknown	T6SS1 & T6SS2 are conserved; T6SS3 & T6SS4 are rare	Tight for T6SS1 & T6SS2; Unknown for T6SS3 & T6SS4	T6SS1: warm (30 °C), seawater-like conditions; T6SS2: low salinity	T6SS1: eliminates *V. cholerae* and other Proteobacteria to promote niche dominance; T6SS2: aids in host adhesion and autophagy
*V. coralliilyticus*	Two (T6SS1 & T6SS2)	T6SS1: antibacterial; T6SS2: anti-eukaryotic	Both conserved across studied strains	Tight for both T6SSs	Both activated under elevated temperatures (e.g., heat waves)	T6SS1: kills coral symbionts and other *Vibrio* spp. facilitating coral host colonization; T6SS2: damages coral tissue and causes mortality in *Artemia*
*V. fischeri*	Two (T6SS1 & T6SS2)	T6SS1: unknown T6SS2: antibacterial	T6SS1 conserved; T6SS2 is strain-specific (~70% of strains)	T6SS1 unknown; T6SS2 is tightly regulated	Viscosity (host mucus-like); acidic/neutral pH (host crypts)	T6SS1: unknown; T6SS2: eliminates inter- and intrabacterial competitors during colonization of *Euprymna* squid host

## Data Availability

No new data were created or analyzed in this study.

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
