# Peer review of "The Competitive Edge: T6SS-Mediated Interference Competition by Vibrionaceae Across Marine Ecological Niches"

_microorganisms, 2025, doi:10.3390/microorganisms13061370_

Round 1
Reviewer 1 Report
Comments and Suggestions for Authors
The manuscript reviewed the T6SS-mediated interference competition by Vibrionaceae across marine ecological niches. The ecological significance of T6SS-mediated interference competition by members of the Vibrionaceae family across a range of marine habitats that include free-living microbial communities and host-associated niches, and the ecological impact of T6SS-mediated competition in modulating biofilm community structure and promoting horizontal gene transfer within those complex microbial populations were examined. This review is well organized and written. The review topic is very suitable for this journal, and has high reference value for researchers in the fields. I recommend it to be published after a minor revision.
- Section 3.1 General Background. The abbreviation T6SS has been defined above.
- Section 4 T6SS in Vibrio: Impact on Marine Free-Living and Host-Associated Microbial Communities. The author reviewed T6SS in different Vibrio species, but there is too much content in this section. The author are suggested to focus on summarizing and comparing the similarities and differences between different bacterial strains.
Author Response
The manuscript reviewed the T6SS-mediated interference competition by Vibrionaceae across marine ecological niches. The ecological significance of T6SS-mediated interference competition by members of the Vibrionaceae family across a range of marine habitats that include free-living microbial communities and host-associated niches, and the ecological impact of T6SS-mediated competition in modulating biofilm community structure and promoting horizontal gene transfer within those complex microbial populations were examined. This review is well organized and written. The review topic is very suitable for this journal, and has high reference value for researchers in the fields. I recommend it to be published after a minor revision.
- Section 3.1 General Background. The abbreviation T6SS has been defined above.
- Section 4 T6SS in Vibrio: Impact on Marine Free-Living and Host-Associated Microbial Communities. The author reviewed T6SS in different Vibrio species, but there is too much content in this section. The author are suggested to focus on summarizing and comparing the similarities and differences between different bacterial strains.
We have addressed these comments and made the appropriate changes in the new version of this manuscript.
Reviewer 2 Report
Comments and Suggestions for Authors
This review article is regarding the prevalence of T6SS-mediated interference competition by Vibrionaceae across marine ecological niches and the competitive edge it provides. The organization of this review and all the sub-sections are well defined and discussed.
The text contents of this review justifies the title. Abstract is well formulated and precise. However, a good figure and elaborated table may add value to this manuscript. The current figures are too basic and obvious for T6SS, while specific figures pertaining to the topic covered in this review is needed.
Overall, the text of this manuscript seems a good piece for the scientific readers.
Author Response
This review article is regarding the prevalence of T6SS-mediated interference competition by Vibrionaceae across marine ecological niches and the competitive edge it provides. The organization of this review and all the sub-sections are well defined and discussed.
The text contents of this review justifies the title. Abstract is well formulated and precise. However, a good figure and elaborated table may add value to this manuscript. The current figures are too basic and obvious for T6SS, while specific figures pertaining to the topic covered in this review is needed.
We have addressed these comments and made the appropriate changes. We added a table for more specific informaiton and an additional figure for more specific points on the T6SS in Vibrio.